# Blood Flow Measurements Enable Optimization of Light Delivery for Personalized Photodynamic Therapy

**DOI:** 10.3390/cancers12061584

**Published:** 2020-06-15

**Authors:** Yi Hong Ong, Joann Miller, Min Yuan, Malavika Chandra, Mirna El Khatib, Sergei A. Vinogradov, Mary E. Putt, Timothy C. Zhu, Keith A. Cengel, Arjun G. Yodh, Theresa M. Busch

**Affiliations:** 1Department of Radiation Oncology, Perelman School of Medicine, University of Pennsylvania, Philadelphia, PA 19104, USA; yihong.ong@pennmedicine.upenn.edu (Y.H.O.); joann.miller2@pennmedicine.upenn.edu (J.M.); minyuan@pennmedicine.upenn.edu (M.Y.); timothy.zhu@pennmedicine.upenn.edu (T.C.Z.); cengel@pennmedicine.upenn.edu (K.A.C.); 2Department of Physics and Astronomy, School of Arts and Sciences, University of Pennsylvania, Philadelphia, PA 19104, USA; malavika.chandra@gmail.com (M.C.); yodh@physics.upenn.edu (A.G.Y.); 3Department of Biochemistry and Biophysics, Perelman School of Medicine, University of Pennsylvania, Philadelphia, PA 19104, USA; elmirna@pennmedicine.upenn.edu (M.E.K.); vinograd.upenn@gmail.com (S.A.V.); 4Department of Chemistry, School of Arts and Sciences, University of Pennsylvania, Philadelphia, PA 19104, USA; 5Department of Biostatistics, Epidemiology & Informatics, Perelman School of Medicine, University of Pennsylvania, Philadelphia, PA 19104, USA; mputt@pennmedicine.upenn.edu

**Keywords:** photodynamic therapy, vascular response, Photofrin^®^, hemodynamic, perfusion, blood flow monitoring, light modulation, diffuse correlation spectroscopy, phosphorescence quenching, Oxyphor

## Abstract

Fluence rate is an effector of photodynamic therapy (PDT) outcome. Lower light fluence rates can conserve tumor perfusion during some illumination protocols for PDT, but then treatment times are proportionally longer to deliver equivalent fluence. Likewise, higher fluence rates can shorten treatment time but may compromise treatment efficacy by inducing blood flow stasis during illumination. We developed blood-flow-informed PDT (BFI-PDT) to balance these effects. BFI-PDT uses real-time noninvasive monitoring of tumor blood flow to inform selection of irradiance, i.e., incident fluence rate, on the treated surface. BFI-PDT thus aims to conserve tumor perfusion during PDT while minimizing treatment time. Pre-clinical studies in murine tumors of radiation-induced fibrosarcoma (RIF) and a mesothelioma cell line (AB12) show that BFI-PDT preserves tumor blood flow during illumination better than standard PDT with continuous light delivery at high irradiance. Compared to standard high irradiance PDT, BFI-PDT maintains better tumor oxygenation during illumination and increases direct tumor cell kill in a manner consistent with known oxygen dependencies in PDT-mediated cytotoxicity. BFI-PDT promotes vascular shutdown after PDT, thereby depriving remaining tumor cells of oxygen and nutrients. Collectively, these benefits of BFI-PDT produce a significantly better therapeutic outcome than standard high irradiance PDT. Moreover, BFI-PDT requires ~40% less time on average to achieve outcomes that are modestly better than those with standard low irradiance treatment. This contribution introduces BFI-PDT as a platform for personalized light delivery in PDT, documents the design of a clinically-relevant instrument, and establishes the benefits of BFI-PDT with respect to treatment outcome and duration.

## 1. Introduction

In photodynamic therapy (PDT), light, photosensitizer and oxygen interact to produce tissue-damaging reactive oxygen species (ROS). PDT is directly cytotoxic to treated cells; moreover, it can promote antitumor immunity and damage tumor vasculature [1,2]. Vascular damage is often beneficial to achieving a complete response, but the timing of vascular damage relative to the period of light delivery is critical [3,4]. Functional deterioration of tumor blood vessels after light delivery is favorable when it deprives remaining tumor cells of oxygen and nutrients. Conversely, therapeutic effects can be compromised when vascular damage manifests as temporary ischemia during light delivery; under these circumstances, a resultant decrease in oxygen supply during PDT can reduce ROS generation and limit cell kill. Subsequently, when tumor blood flow recovers after PDT, the surviving treatment-spared tumor cells can proliferate to result in tumor regrowth. For these reasons, the extent and time-course of PDT-induced vascular damage can critically impact therapeutic effect.

The ability to monitor and modulate tumor blood flow during illumination offers a means for real-time optimization of PDT delivery. PDT is well known to induce fluctuations in tumor blood flow during illumination across various treatment protocols [4,5,6,7,8,9,10,11]. Khurana et al. [5] used Doppler optical coherence tomography to observe a marked increase in blood flow velocity at the start of illumination and, attributed this to a narrowing of blood vessel lumens. Similarly at the start of illumination, Yu et al. [11] observed an increase and then a sharp decline in blood flow using optical diffuse correlation spectroscopy. The rate of this decline correlated strongly with treatment outcome. Specifically, tumors with rapid decrease in blood flow experienced shorter regrowth delays. The relative changes in blood flow and the tissue hemoglobin oxygen saturation of tumors after PDT were also predictive of outcome. Nevertheless, blood flow during PDT is a more appealing clinical metric, because it enables outcome to be predicted using data obtained during treatment. 

Vascular and cytotoxic responses to PDT are affected by fluence rate in the treated tissue, which is controlled by the choice of irradiance, i.e., the incident light fluence rate on the tissue in the case of surface illumination [12,13,14]. In protocols dominated by anti-vascular mechanisms, nonreversible ischemia may build within the tumor over the course of illumination. Yet, for other PDT protocols, efficacy is determined by the combined effects of vascular damage and direct tumor cell damage. Lower irradiances are beneficial under these circumstances, and this benefit is often attributed to the conservation of blood flow during illumination and a reduction in the rate of oxygen consumption by PDT. Consequently, longer illumination times at low irradiance can favor long term outcomes by enhancing direct PDT cytotoxicity and vascular damage after light delivery [10,12,15,16,17,18,19]. By contrast, higher irradiances may compromise direct tumor cell kill by inducing reversible blood flow stasis during light delivery and lowering ROS production as a result of tissue hypoxia, [20,21]. Fractionated illumination with short on-off cycles or two-fold illumination with a single long dark interval also appears to improve PDT effectiveness [22,23,24,25,26,27,28,29,30,31]. In this case, treatment efficacy varies with the lengths of dark intervals and first light fraction, as well as the irradiance level. Unfortunately, despite evidence that low irradiance and light fractionation improve PDT outcomes, such approaches can be clinically undesirable due to lengthy treatment times. 

Here we introduce a novel scheme, blood-flow-informed PDT (BFI-PDT), to personalize PDT delivery. BFI-PDT dynamically adjusts the length of light/dark fractions and/or choice of irradiance based on real-time measurement of PDT-induced vascular response. We hypothesize that BFI modulation of light delivery improves PDT efficacy compared to standard treatment with continuous high irradiance, while limiting treatment times to less than those needed for continuous illumination at low irradiance. We explore these ideas using two preclinical murine tumor models, radiation-induced fibrosarcoma (RIF) and malignant mesothelioma (AB12). The hemodynamic characteristics and tumor responses achieved by three standard and two BFI-PDT treatment schemes are studied. Noninvasive measurement of relative blood flow (rBF) in tumor is measured by diffuse correlation spectroscopy (DCS). This technology has been validated and employed for monitoring of blood flow in brain [32,33,34], skeletal muscle [35,36], and tumors [37,38], as well as for determination of vascular response during PDT [1,7,9,11,39,40]. DCS is sensitive to flow in tumor microvasculature, i.e., tumor arterioles, capillaries, and venules, and it can be used in a noncontact configuration that permits continuous monitoring during PDT without interfering with treatment light delivery. In the preclinical murine models, we demonstrate the ability to measure and modulate tumor blood flow via real-time BFI light delivery. For these tumors BFI-PDT minimizes the duration of illumination and is both safe and effective. 

## 2. Materials and Methods

### 2.1. Tumor Models/PDT

RIF and AB12 tumors were grown in C3H or BALB/c mice (Charles River Laboratories, Wilmington, MA, USA), respectively. A total of 3 × 10^5^ RIF or 1 × 10^6^ AB12 cells were injected intradermally over the right shoulder or the flank of the mice, respectively. The animals were entered in studies ~one week later with tumor diameters of ~5 mm. The treatment area was depilated (Nair hair removal lotion, Church & Dwight Co., Inc., Ewing, NJ, USA), and Photofrin^®^ was injected 20–24 h before illumination (tail vein, 5 mg/kg) to allow for its accumulation in tumor tissues [41]. Photofrin^®^ distributes to both malignant cells and blood vessels in tumors, and with longer incubations it localizes mainly to organelle membranes such as the mitochondria, endoplasmic reticulum and Golgi complex [42,43]. 

Treatment light from a 630-nm laser (Biolitec AG., A-1030, Vienna, Austria) was collimated using a microlens-tipped fiber (CardioFocus, Norton, MA, USA). It was delivered over a 1 cm diameter field centered on the tumor to a total fluence of 135 Jcm^−2^, at a high irradiance of 150 mWcm^−2^, a low irradiance of 25 mWcm^−2^, or a combination of these irradiances as a function of the treatment scheme. It should be noted that external beam PDT induces no to minimal tissue heating (2–3 °C) at irradiances ≤150 mWcm^−2^ [44]. Mice were anesthetized using ~1.5% isoflurane, while a heating pad maintained body temperature. 

Tumor response was quantified as the number of days after PDT for tumor regrowth to a volume of 400 mm^3^ (i.e., time-to-400mm^3^). Tumor volume (V) was calculated as V = π/6 × width^2^ × length. An absence of tumor burden at 90 days after PDT was defined as a complete response. Animal studies were approved by the IACUC of the University of Pennsylvania and animal facilities accredited by AAALAC under protocol #803526 (approved on 3/24/2011, latest renewal 2/12/2020).

### 2.2. Diffuse Correlation Spectroscopy

Diffuse correlation spectroscopy (DCS) derives information about tissue blood flow by quantifying the temporal intensity fluctuation of multiple scattered light. It employs coherent near-infrared (NIR) light that travels diffusively and scatters within tissue before emerging from the tissue surface. Each scattering event alters the phase of the scattered light field; eventually, multiple light fields travel from the light source to the tissue surface, each with different phases. These fields interfere constructively or destructively to create a speckle pattern. The movements of red blood cells in tissues alter the scattered light phases and cause the speckle pattern to vary in time. Thus, the temporal intensity fluctuations of the detected light are sensitive to the motions of red blood cells in the tissue microvasculature. The temporal decay of the autocorrelation function (the DCS signal) therefore provides a direct measure of blood flow. A tumor blood-flow index (BFi) is obtained by fitting the measured intensity autocorrelation function to the solution of the correlation diffuse equation in the semi-infinite homogeneous medium geometry [45,46]. 

#### 2.2.1. DCS Instrumentation

The DCS instrumentation includes a coherent laser source (785 nm, 80 mW, CrystaLaser Inc., Reno, NV, USA) operating in continuous mode, and a fiber bundle consisting of 4 single mode detector fibers located at 0.2 cm around a central multimode source fiber. The fibers were placed at the detector plane at the back of a camera while the camera lens delivered excitation light and collected reflected diffuse light from the tumor surface. Four single photon-counting avalanche photodiodes (SPCM-4AQC, Excelitas, Waltham, MA, USA) were employed in parallel for detection of the diffuse light. Typically, average photon penetration depth into tissue is one-third to one-half of the source-detector separation. Therefore, when the excitation light was focused onto the center of tumor surface, light detected by the detector fibers is primarily sensitive to superficial tumor tissue (~0.7–1 mm depth) spanned between the source-detector pairs and not the underlying normal tissue. Bandpass optical filters are used to attenuate the treatment light and prevent the detectors from saturation. The detected intensity autocorrelation functions were computed using a four-channel autocorrelator board (Correlator.com, Bridgewater, NJ, USA) at ~0.34 Hz measurement rates. DCS data for one frame were averaged over all four detector channels to represent the rBF in the bulk tumor tissue. This setup enabled us to monitor blood flow during PDT by permitting unobstructed illumination with the treatment light at a small angle to the tissue surface.

#### 2.2.2. Tumor Blood Flow Monitoring 

Tumor blood flow (BF) was monitored continuously by DCS beginning ~10 min before light delivery (the baseline period) until ~70 min after its completion. Figure 1 illustrates the light delivery system for BFI-PDT. A digital attenuator (DA-100-5-600/700-400/430-M/35-SP-HP, OZ Optics Ltd., Ottawa, ON, Canada) was positioned in-line with the output fiber from the treatment laser and used to alter the irradiance incident on the tumor. A DCS module, with a noncontact probe, was mounted above the targeted tissue to monitor tumor blood flow. Relative blood flow (rBF) was calculated by normalizing the blood flow index (denoted BF_i_) measured at time *t* (i.e., BF_i_(*t*)) to the baseline flow measurements (BF_i_(0)); $rBF\left(t\right)=BF_{i}\left(t\right)/BF_{i}\left(0\right)\times 100\%$. The percent change in rBF per minute (i.e., %rBFmin^−1^) or the slope of rBF was computed in real time by fitting a linear regression model to 72 rBF readings (~3.5 min of data). These parameters provided the source data for defining BFI-PDT light delivery and for in situ decisions about how light irradiances should be varied. In practice, DCS is susceptible to motion artifacts due to sudden movements. Random spikes in blood flow may occur, for example, due to occasional deep breathing or twitching of a muscle during light delivery. Therefore, to avoid false triggers of irradiance changes, the rBF data train was smoothed with a window of 10 data points (~30 s). 

### 2.3. Illumination Schemes

Three standard PDT (1, 2 and 3) and two BFI-PDT illumination schemes (4 and 5) were used to deliver PDT to a total fluence of 135 Jcm^−2^. For each scheme, 10–12 mice were treated.

150 mWcm^−2^-continuous: Continuous illumination at high irradiance of 150 mWcm^−2^ for 15 min.25 mWcm^−2^-continuous: Continuous illumination at low irradiance of 25 mWcm^−2^ for 90 min.150 mWcm^−2^-fractionated: 150 mWcm^−2^ in equal intermittent intervals of 30 s light-on and 30 s light-off for a total of 30 min.Blood-flow-informed-irradiance (BFI-Irrad): Continuous illumination was initially 150 mWcm^−2^, but illumination was cyclically decreased to 25 mWcm^−2^ and returned to 150 mWcm^−2^ in response to the blood flow monitoring parameters. Treatment time was adjusted to deliver a total fluence of 135 Jcm^−2^ (between 15 and 90 min).Blood-flow-informed-fractionated (BFI-Frac): Fractionated illumination was initiated at 150 mWcm^−2^, but illumination was intermittently discontinued (light-off, 0 mWcm^−2^) in response to blood flow monitoring. Treatment time was adjusted to deliver a total fluence of 135 Jcm^−2^, which was reached within 90 min in the current investigations. Note, this guidance platform requires a maximum treatment time to be established irrespective of whether or not a total fluence of 135 Jcm^−2^ is achieved because the light can remain “off” for extended periods of time if blood flow recovery is slow.

### 2.4. In Vivo/In Vitro Clonogenic Assay

Tumor-bearing animals were treated with PDT or as controls. At the indicated timepoints, animals were euthanized, and tumors were excised, weighed, minced and enzymatically digested using a technique described previously [19,47]. In brief, the minced tumor was suspended in a trypsinizing flask containing 3000 units deoxyribonuclease (Sigma-Aldrich, St. Louis, MO, USA), 2000 units collagenase (Sigma-Aldrich), and 3 mg protease (Sigma-Aldrich) dissolved in 12 mL of Hanks’ balanced salt solution. After digestion, cells were plated on 100 mm tissue culture dishes in triplicate, and after a ~10-day incubation (37 °C in 95% air/5% CO_2_) colonies were fixed, stained (2.5 mg/mL methylene blue in 30% alcohol), and counted. The number of clonogenic cells per gram was calculated as the number of cells per gram of tumor multiplied by the ratio of the number of colonies to the number of cells plated.

### 2.5. Tumor Oxygenation Measurements

Phosphorescence lifetime-based measurements of tumor oxygenation were performed as described previously [39,48,49]. Oxyphor R4 (30 µL, 20 mM), whose absorption spectrum does not overlap with the emission band of the treatment laser, was injected intra-tumorally ~24 h prior to measurement. Excitation of the phosphorescence was performed using a light-emitting diode (λ_max_ = 523 nm), and the phosphorescence was detected using an avalanche photodiode (APD) through a long-pass filter (710 nm). Phosphorescence lifetime oximetry permits absolute measurement of tumor oxygen partial pressure. The technique is based on variations in the probe phosphorescence decay time due to quenching of the probe triplet state by oxygen. The measurements are thus unaffected by the concentration of the probe, the excitation light intensity, and signal collection efficiency. 

### 2.6. Statistical Analysis

Differences in the distribution of flow reduction between each of the modalities were compared using a Kruskall Wallis test, followed by pairwise Wilcoxon Rank Sum tests. Differences in log-transformed tumor clonogenicity and ΔrBF were assessed using a one-way ANOVA. To account for repeated measurements on the same animal, differences between mean pO_2_ were analyzed using a mixed effects model. Median time-to-400 mm^3^ was estimated using the Kaplan-Meier method. In order to weight earlier failure times more strongly, differences in the time-to-400 mm^3^ among groups were assessed using Gehan-Wilcoxon tests [50]. The proportion of animals with a complete response was estimated and the exact 95% confidence interval (CI) reported. The association between regrowth and either treatment group or flow reduction rate was assessed using a Cox model. Statistical analyses were carried out in R (3.6.1) with the package *survminer* for pairwise comparisons of the time-to-event data. The family-wise error rate was maintained at 0.05 using either a Holm-Bonferroni correction for multiplicity (flow reduction rate, pO_2_, or time-to-400 mm^3^) or Tukey’s Honest Significance Difference (clonogenicity, ΔrBF).

## 3. Results

### 3.1. Irradiance Alters Tumor Blood Flow During PDT

Initial studies characterized rBF in tumors treated with standard PDT. Figure 2 depicts representative traces of rBF during light delivery for PDT with (a) high irradiance-continuous (150 mWcm^−2^ in red), (b) low irradiance-continuous (25 mWcm^−2^ in yellow) and (c) high irradiance-fractionated illumination (150 mWcm^−2^ light-on intervals in red). 

For all standard illumination schemes, rBF increased at the start of light delivery (Figure 2a–c). At high irradiance, after a rapid increase, rBF declined steeply. By contrast, at low irradiance, after a more gradual initial increase, rBF declined more slowly. These changes in rBF during PDT are characterized by rBF_max(initial)_ and rBF_min(initial)_, defined as the respective maximum and minimum flow associated with this first rise and fall (Figure 2a–c). A flow reduction rate was calculated for each mouse as the slope of the line segment between rBF_max(initial)_ and rBF_min(initial)_ (depicted by the dotted line). During continuous irradiance at 25 mWcm^−2^ the median flow reduction rate was 9.3 %rBFmin^−1^, a value significantly smaller than the median flow reduction rate of 23 %rBFmin^−1^ for 150 mWcm^−2^-continuous (*p* < 0.001) and 25 %rBFmin^−1^ for 150 mWcm^−2^-fractionated (*p* = 0.0016) (Figure 2d and Table 1). 

Figure 2e shows a Kaplan-Meier curve for time-to-400 mm^3^ with groups defined by tertiles of flow-reduction. As in previous studies [11], larger flow reduction rates were associated with worse treatment outcome (*p* < 0.001 ). In a Cox model, the instantaneous ‘risk’ of regrowth to 400 mm^3^ increased by a factor of 2.1 (95% confidence interval, 1.6, 2.7) per 10% increase in flow reduction rate (*p* < 0.001). In a Cox model including only the treatment groups, there was a significant association with time time-to-400 mm^3^ (*p* = 0.05), but once flow reduction rate was added to the model, treatment group was no longer statistically significant (*p* = 0.87). This suggests that a considerable portion of the treatment effect was mediated through flow reduction rate. 

The above results suggest irradiance affects flow reduction rate, and that a larger flow reduction rate is associated with poorer response across a range of standard treatment conditions (i.e., high and low irradiance, continuous and fractionated illumination). Furthermore, these findings suggest flow reduction rate could serve as an optimization criterion to personalize PDT for individual animals. Our approach proposed monitoring blood flow during PDT and altering irradiance in real time in response to changes in blood flow. With this scheme, we hoped to maintain favorable flow reduction rates during light delivery and thus improve PDT outcomes.

### 3.2. BFI-PDT Alters Flow Dynamics during Illumination Compared to Standard PDT

BFI-PDT was studied with schemes of either BFI-Irrad or BFI-Frac as defined in Figure 3a. The delivered light fluence, accumulated treatment time, relative tumor blood flow, and flow reduction rate were monitored continuously throughout light delivery. BFI-Irrad utilized either a time-conserving irradiance of 150 mWcm^−2^ or flow-conserving 25 mWcm^−2^. BFI-Frac employed either an irradiance of 150 mWcm^−2^, or when the laser was paused, an irradiance of 0 mWcm^−2^. 

During PDT, relative blood flow (rBF(*t*)) and rBF slope were collected as a function of time. For both treatment schemes, PDT commenced with light delivery at 150 mWcm^−2^. Irradiance was attenuated to 25 mWcm^−2^ for BFI-Irrad (exemplified in Figure 3b), or to 0 mWcm^−2^ for BFI-Frac (exemplified in Figure 3c), when the reduction in rBF during PDT was more rapid than a cutoff value of 10% per minute (i.e., when the “time-derivative” of rBF(*t*) was less than or equal to −10% rBFmin^−1^). This cutoff value was selected based on our previous work [11] where we observed rapid tumor regrowth after PDT (time-to-400 mm^3^ of <10 days) when flow reduction rate was higher than 10 %rBFmin^−1^, and substantial delay in tumor regrowth when flow reduction rate was less than or equal to 10 %rBFmin^−1^. If rBF(*t*) recovered to a value above its level during the baseline period (rBF_baseline_), then the irradiance returned to 150 mWcm^−2^. At each subsequent instance of a flow decrease of more than 10% rBFmin^−1^, the irradiance was again attenuated to 25 mWcm^−2^ (BFI-Irrad) or 0 mWcm^−2^ (BFI-Frac), and then, the irradiance was returned to 150 mWcm^−2^ when rBF(*t*) recovered to above rBF_baseline._ Light was delivered until a total fluence of 135 Jcm^−2^ was reached, or, in the case of BFI-Frac, until a maximum treatment time of 90 min (including light-off time). Appendix A additionally illustrates how temporal fluctuations in rBF (Appendix A) associate with the slope of its change (Appendix A); BFI-PDT attenuated the irradiance of light delivery when this slope decreased at a rate greater than 10% rBFmin^−1^.

Several features distinguish the flow reduction rate for BFI-PDT compared to standard PDT. As shown in Figure 2a–c, rBF_min(initial)_ for standard PDT generally represents the global minimum in rBF (rBF_min(global)_) for the entire treatment. By contrast, during BFI-PDT, modulation of light delivery interrupts and reverses PDT-induced decreases in rBF. Consequently, rBF_min(global)_ rarely corresponds with rBF_min(initial)_ for BFI-PDT. The rBF_min(global)_ during BFI-PDT often occurs later during treatment, i.e., after illumination has been attenuated or paused at least once per the rules of the BFI platform. Furthermore, maximum rBF during standard PDT is generally the first peak in rBF after the start of illumination (i.e., rBF_max(initial)_), which differs from BFI-PDT because the BFI platform promotes blood flow recovery that may lead to subsequent peaks higher than rBF_max(initial)_. These differences in blood flow trends during BFI-PDT versus standard PDT provide hemodynamic evidence of a modified vascular response during illumination.

### 3.3. BFI-PDT Decreases Blood Flow Reduction Rate during Light Delivery, While Shortening Treatment Time

Table 1 summarizes the hemodynamic effect (flow reduction rate) and treatment parameters (duration of exposure to each fluence rate) for RIF tumors treated with standard PDT or BFI-PDT. We note that BFI-Irrad and BFI-Frac induced multiple peaks and troughs in rBF during treatment; the hemodynamic effect was quantified as a single overall flow reduction rate, calculated from the slope of decrease between rBF_max(initial)_ and rBF_min(global)_, as depicted by the red dotted lines in Figure 3b,c. Median overall flow reduction rate for BFI-Irrad was 5.2% rBFmin^−1^ and for BFI-Frac was 10% rBFmin^−1^. These values were compared to those of standard conditions. Overall flow reduction rates for BFI-Irrad were lower than those for each of the standard PDT conditions with continuous illumination (*p* ≤ 0.001). Overall flow reduction rates for BFI-Frac were lower than for 150 mWcm^−^**^2^** using either continuous or fractionated light (*p* < 0.001 in each case) but did not differ from 25 mWcm^−^**^2^** continuous (*p* = 0.63).

We next investigated how BFI-PDT affects treatment length. A total of 90 min was required to deliver 135 Jcm^−2^ at 25 mWcm^−^**^2^**-continuous. Median treatment length for BFI-Irrad was 53 min, about 59% of the time required for 25 mWcm^−^**^2^**-continuous. During BFI-Irrad, total fluence was divided about equally between 150 mWcm^−2^ and 25 mWcm^−2^, corresponding to ~15% of the illumination time at 150 mWcm^−2^ and ~85% of the time at 25 mWcm^−2^. With a median treatment time of 61 min, BFI-Frac took twice as long as for treatment with standard 150 mW/cm^2^-fractionated (30 min), but less time than treatment with 25 mWcm^−^**^2^**-continuous (90 min). Unlike BFI-Irrad, BFI-Frac incorporates light-off periods. For BFI-Frac, there is no pre-specified maximum time for completion of treatment; if rBF remains below the pre-PDT baseline value for extensive periods, the desired fluence cannot be delivered in a reasonable time. For these reasons, BFI-Irrad is more clinically relevant than BFI-Frac.

### 3.4. BFI-PDT Alters Mechanisms of PDT Effect on Vascular Damage

Tumor vascular perfusion has been studied using techniques such as laser Doppler [5], power Doppler ultrasound imaging [11] and various histological and other assays [20,51,52]. Here, the mechanistic effect of BFI-PDT on vascular damage was studied by assessing ΔrBF at 1 h after PDT measured by DCS. We have previously shown that DCS measurements after PDT aligned with those measured by power Doppler [11]. This demonstrates that we are measuring decreases in perfusion as an indication of vascular damage. Vascular congestion develops within 3 h after treatment of RIF tumors with PDT [51], suggesting that this congestion contributes to the detected decreases in blood flow.

Representative traces show rBF through 1 h after PDT for each treatment scheme (Appendix A). ΔrBF was defined as the calculated difference between rBF at 1 h after PDT and rBF at baseline, representing the change in vascular function between these two timepoints. As previously shown [53], lower irradiance led to more vascular damage after PDT than did higher irradiance. Among the standard conditions, 25 mWcm^−2^-continuous (mean ΔrBF of −46%; *p* < 0.001 and 150 mWcm^−^**^2^**-fractionated (−32%; *p* = 0.018) both resulted in substantially more damage to tumor vasculature than 150 mWcm^−2^-continuous (+25%). For BFI-PDT, both BFI-Irrad (mean −31%; *p* = 0.016) and BFI-Frac (−44%; *p* = 0.002) led to more vascular damage after PDT than 150 mWcm^−^**^2^**-continuous. In contrast, each BFI-PDT scheme resulted in similar mean ΔrBF compared to 25 mWcm^−^**^2^**-continuous or 150 mWcm^−^**^2^**-fractionated PDT (*p* > 0.05 in each comparison). Thus, PDT produces more vascular damage than the comparative standard treatment with 150 mWcm^−2^-continuous.

The effect of BFI-PDT on direct cytotoxicity was assessed by clonogenic assay. Mice were euthanized for tumor excisions at timepoints A-E as indicated in Figure 4a. Tumor clonogenicity (log transformed) for each treatment condition/timepoint is presented in Figure 4b. Compared to control (timepoint A), PDT with 150 mWcm^−2^-continuous did not reduce tumor clonogenicity. The number of clonogenic cells/g (mean ± SD) was 1.16 ± 0.35 × 10^8^ for controls and 1.06 ± 0.63 × 10^8^ at the conclusion PDT (timepoint B). Even when tumors were left for an additional 45 min (a time lapse of 60 min from the start of illumination; timepoint C), treatment with 150 mWcm^−2^-continuous did not yield cytotoxicity (mean = 1.61 ± 1.4 × 10^8^). In contrast, compared to controls, both BFI-Irrad and BFI-Frac dramatically reduced the number of clonogenic cells/g to 0.18 ± 0.15 × 10^8^ (*p* < 0.001) and 0.13 ± 0.11 × 10^8^ (*p* < 0.001), respectively, for tumor assayed immediately upon treatment conclusion (timepoints D and E for BFI-Irrad and BFI-Frac, respectively). Because treatment with BFI-Irrad and BFI-Frac required 53 to 61 min, tumor clonogenicity was compared to that at 60 min after the start of 150 mWcm^−^**^2^**-continuous (timepoint C). At this timepoint, both BFI-Irrad (*p* < 0.001) and BFI-Frac (*p* < 0.001) created an additional ~1 log_10_ of cell kill compared to 150 mWcm^−^**^2^**-continuous. Thus, BFI-PDT was consistently more cytotoxic to tumor cells than 150 mWcm^−^**^2^**-continuous standard PDT.

Conservation of tumor blood flow by BFI-PDT is expected to maintain tumor oxygenation during light delivery and thus induce more cytotoxicity compared to standard PDT. To understand the effect of BFI-PDT on tumor oxygenation, partial oxygen tension was assessed by phosphorescence lifetime oximetry for 1-min periods before and after PDT (indicated by the dark bands on Figure 4a). Across all animals, mean tumor oxygenation at baseline was 32.8 mmHg (Figure 4c). After control (light only) treatment, mean oxygenation levels (32.6 mmHg) were indistinguishable from baseline values. Compared to control, mean pO_2_ declined in all experimental groups immediately after illumination (*p* < 0.001 for each group). At the conclusion of 150 mWcm^−2^-continuous illumination, mean tumor oxygenation decreased sharply to a mean of 13.3 mmHg, in contrast to more modest declines to means of 26.7 mmHg and 25.5 mmHg respectively for BFI-Irrad and BFI-Frac (*p* < 001 for each compared to 150 mW cm^−2^-continuous). Interestingly, tumor oxygenation recovered over the 45 min after 150 mWcm^−2^-continuous to a mean pO_2_ of 26.2 mmHg at timepoint C (*p* < 0.001 compared to timepoint B). This reoxygenation likely reflected recovery of tumor blood flow after PDT, as shown in Appendix A. The acute hypoxia induced by illumination with 150 mWcm^−2^-continuous could therefore limit direct cytotoxicity to tumor cells, and spare tumor vasculature.

### 3.5. BFI-PDT Improves Therapeutic Outcome

Tumor response studies were performed to assess the overall therapeutic value of BFI-PDT (Figure 5). For RIF-bearing mice treated with 150 mWcm^−2^-continuous, the median time-to-400 mm^3^ was 11.0 days compared to 19.6 days for 25 mWcm^−2^-continuous (*p* = 0.031 versus 150 mWcm^−2^-continuous) and 18 days for 150 mWcm^−2^-fractionated (*p* = 0.048 versus 150 mWcm^−2^-continuous (Figure 5a and Appendix A)). No animals at 150 mWcm^−2^-continuous achieved a complete response compared to 30% (95% confidence interval (CI) of 7–65%) at 25 mWcm^−2^-continuous and 22% (95% CI of 2–48%) at 150 mWcm^−2^-fractionated. Importantly, however, standard fractionation was also associated with morbidity that was not found with either of the continuous treatment schemes. Fractionation promoted high levels of edema and culminated in mortality within several days of PDT for 25% (95% CI of 5–57%) of animals.

Both BFI-Irrad and BFI-Frac were more effective than 150 mWcm^−2^-continuous at inducing a tumor response. For BFI-Irrad, the median time-to-400 mm^3^ was 29 days (*p* = 0.006 versus 150 mWcm^−2^-continuous) and 40% (95% CI of 12–74.5%) of animals exhibited a complete response (Figure 5b and Appendix A). No differences in tumor response were observed for BFI-Irrad versus 25 mWcm^−2^-continuous (*p* = 0.88), but the required treatment time was ~40% shorter for BFI-Irrad than 25 mWcm^−2^-continuous (see Table 1). 

For BFI-Frac, the median time-to-400 mm^3^ was 39 days and 56% (95% CI of 18–91%) achieved a complete response (*p* < 0.002 versus 150 mWcm^−2^-continuous). No differences in tumor response were observed for BFI-Frac versus 25 mWcm^−2^-continuous (*p* = 0.45). BFI-Frac appeared less acutely toxic than standard 150 mWcm^−2^-fractionated illumination. With BFI-Frac, 10% (95% CI of 0–45%) of animals experienced a non-acute death (>1 week after PDT), compared to 25% (95% CI of 5–57%) acute deaths within a week of standard 150 mWcm^−2^-fractionation. 

The benefit of PDT with BFI-PDT was confirmed in the AB12 tumor model (Figure 5c and Appendix A). In this model, complete response occurred in 33% (95% CI of 7–71%) for 150 mWcm^−2^-continuous, 89% (95% CI of 52–100%) for 25 mWcm^−2^-continuous and in 100% (95% CI of 69–100%) for BFI-Irrad. The median survival time was 20 days for 150 mWcm^−2^-continuous and exceeded 90 days for both 25 mWcm^−2^-continuous (*p* = 0.021) and BFI-Irrad (*p* = 0.007). Survival times did not differ significantly for 25 mWcm^−2^-continuous versus BFI-Irrad (*p* = 0.29). The median length of treatment with BFI-Irrad was ~60% shorter than that required for 25 mWcm^−2^-continuous. Table 2 summarizes illumination duration and other treatment parameters for AB12 tumors. The median overall flow reduction rate for BFI-Irrad was significantly slower at 6% rBFmin^−1^ compared to 21% rBFmin^−1^ for 150 mWcm^−2^-continuous (*p* = 0.004) or 13% rBFmin^−1^ at 25 mWcm^−2^-continuous (*p* = 0.04). BFI-Irrad also produced more vascular shutdown at 1 h after PDT (−59% ΔrBF) compared to 150 mWcm^−2^-continuous (−11% ΔrBF) (*p* = 0.001). Thus, BFI-Irrad PDT produced similar benefit to tumor vascular damage, complete response, and treatment length in both RIF and AB12 tumors.

## 4. Discussion

The therapeutic effect of PDT depends on the availability of photosensitizer, light and oxygen in the targeted tissue. Besides total fluence-dependency, irradiance or the incident fluence rate in tissue also determines treatment efficacy. Foster et al. [16] demonstrated irradiance effects in PDT of tumor spheroids that were subject to equivalent fluences, delivered at different irradiances. These data found greater cytotoxicity after PDT at lower irradiances. In murine models, lower irradiances are more efficient, requiring lower fluence to achieve the same tumor responses than higher irradiances [53]. Moreover, other in vivo studies [54,55] also demonstrated that lower irradiances improve PDT outcome.

In the present report we observed continuous illumination with 25 mWcm^−2^ to be more effective than 150 mWcm^−2^; however, treatment to a therapeutically relevant fluence of 135 Jcm^−2^ requires substantially longer time at 25 mWcm^−2^ (90 min) than at 150 mWcm^−2^ (15 min). Initially upon illumination, tumor blood flow decreased rapidly at high irradiance, in contrast to more gradual changes at lower irradiance. Rapid decreases could promote hypoxia limiting the therapeutic effect. From these observations, we posited that an interactive approach to light delivery could be guided in real-time by tumor blood flow, utilizing high irradiance during periods of stable blood flow to provide time-efficient light delivery, coupled with low irradiance during periods of declining blood flow to facilitate its recovery. Indeed, results demonstrate that when compared to high irradiance treatment, BFI-PDT interrupts the PDT-induced decrease in blood flow during illumination, conserves tumor oxygenation, increases direct tumor cytotoxicity, and promotes vascular damage after treatment. The tumor damage inflicted by BFI-PDT contributes to significant improvement in long-term therapeutic outcome compared to high irradiance treatment. 

BFI-PDT appears to improve outcome by generating a vascular response similar to that associated with low irradiance PDT. Low irradiance is known to promote vascular shutdown after PDT [10,18]. Additionally, BFI-PDT improves PDT outcome is through an effect on the direct cytotoxicity of PDT to tumor cells. Both BFI-Irrad and BFI-Frac are significantly more cytotoxic to tumor cells than 150 mWcm^−2^-continuous, both immediately after illumination and ~60 min after the start of illumination. Both BFI-PDT conditions preserve tumor oxygenation during treatment compared to 150 mWcm^−2^-continuous. Maintenance of tumor blood flow during light delivery appears to ensure better supply of oxygen, which, in turn, could contribute to production of more cytotoxic ROS and greater tumor damage. Lower irradiance or light-off cycles during BFI-PDT may also reduce photochemical oxygen consumption, further preserving tumor oxygenation during these treatments.

In studies of fractionated PDT, 150 mWcm^−2^ illumination was divided into equal 30- second light-on and light-off intervals, producing treatment outcome similar to that achieved by 25 mWcm^−2^-continuous. However, the benefit of fractionated illumination has been inconsistently demonstrated, with others failing to reveal a treatment benefit [56]. These discrepancies may reflect the lack of an informed approach for fractionation. Choice of irradiance, frequency of light fractions and duration of each pause could each affect treatment efficacy. Moreover, the tumor control provided by fractionated illumination in the present study was accompanied by morbidity, resulting in a 25% acute death rate. This is not unexpected, as others have reported high fluence rate illumination could lead to significant morbidity from the inflammation that it may induce [57,58]. In this study, BFI-Frac guided the insertion of pauses in illumination after rapid decreases in blood flow, allowing treated tissue to re-perfuse before resuming illumination. The observed complete response rate was better in BFI-Frac compared to standard fractionated PDT, and importantly, BFI-Frac did not lead to any acute morbidity. BFI-Frac had one delayed morbidity (4 weeks post-PDT) of unknown cause, whereas no morbidities of any kind were associated with BFI-Irrad.

BFI-Frac however does not appear practical in clinical situations. A rapid decline in tumor blood flow triggers a stop to illumination and thus a pause in treatment. A maximum treatment length cannot be defined in advance and treatment may not even be completed if blood flow remains low for a prolonged period. Based on the currently defined parameters for BFI-PDT, BFI-Irrad appears to be the more valuable treatment platform. However, continued refinement of treatment parameters, such as the choice of treatment irradiances and the threshold rBF slopes that trigger change in irradiances employed for BFI-PDT, are very likely to provide further improvements in response. In this regard, we note that as it is presently applied, the BFI treatment platform demonstrated a capability both to transform noncurative PDT into a treatment with curative potential (in RIF tumors) and to move a potentially curative treatment into a completely curative regimen (in AB12 tumors).

In the present study, we carefully looked at the effect of BFI protocols on major PDT endpoints including tumor oxygenation, vascular damage and tumor regrowth. Future work should look at other subtle changes at the vascular and cellular level, such as the effect of BFI-PDT on vascular permeability. It has been shown that vascular permeability in tumors can be affected by fluence rate modification, for which it peaked at intermediate to low fluence rates and diminished toward very high and very low fluence rates [13]. The enhancement of tumor permeability can be used to increase the deposition and accumulation of drugs into tumors. Understanding how PDT treatments with varying irradiances affect vascular permeability in tumors would be helpful for the optimization of BFI protocols to enhance these effects. Improving PDT-mediated vascular permeabilization could potentially enhance the delivery of chemotherapeutic agents to the tumor, thereby offering added advantage to combination strategies.

## 5. Conclusions

In conclusion, a noninvasive system for real-time monitoring of tumor blood flow during PDT demonstrated personalized light delivery in an automated fashion. In two murine models, PDT with BFI-Irrad achieved a significantly better treatment outcome than PDT using continuous high irradiance. BFI-PDT provided the desired benefit of achieving a therapeutic response that was similar to that of low irradiance PDT but required a much shorter treatment time. Ultimately, this research suggests that the ability to measure and modulate tumor physiologic properties during illumination for PDT will provide a means for personalized delivery in clinical applications.

## Figures and Tables

**Figure 1 cancers-12-01584-f001:**
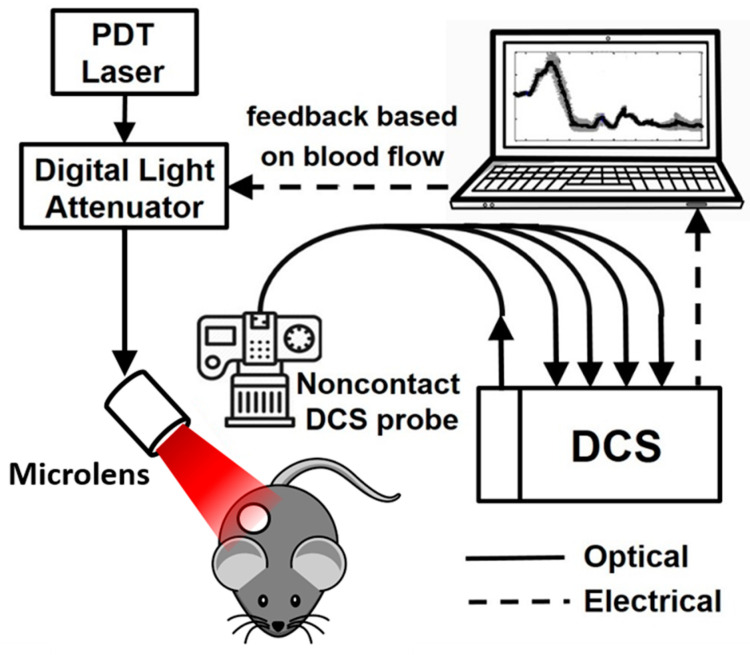
Schematic of blood-flow-informed photodynamic therapy (BFI-PDT) light delivery system.

**Figure 2 cancers-12-01584-f002:**
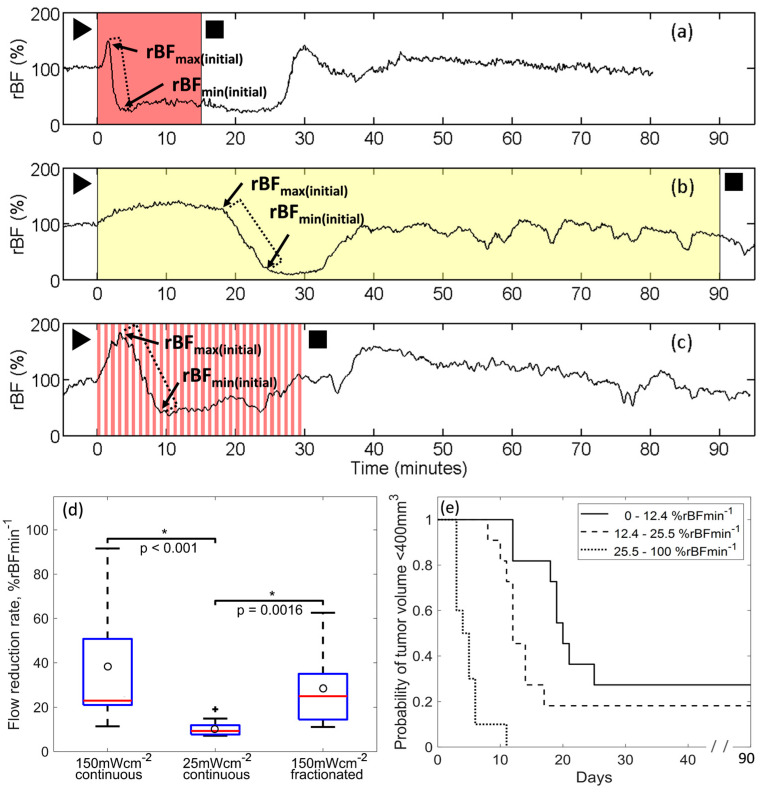
Representative blood flow traces during standard light delivery to radiation-induced fibrosarcoma (RIF) tumors using (**a**) 150 mWcm^−2^-continuous; (**b**) 25 mWcm^−2^-continuous; and (**c**) 150 mWcm^−2^ standard fractionated illumination with 30 second light-on/light-off intervals. Light delivery is shaded in red or yellow for illumination at 150 mWcm^−2^ or 25 mWcm^−2^, respectively. ► and ■ indicate the initiation and completion of light delivery, respectively. rBF_max(initial)_ and rBF_min(initial)_ are the maximum and minimum that define the initial peak and trough in relative blood flow (rBF) during photodynamic therapy (PDT). A dotted bracket on each plot represents the slope of rBF decrease, i.e., flow reduction rate. (**d**) Box plots of blood flow reduction rate for standard PDT treatments (open circles indicate means; *n* = 10–11 mice per group). Statistical differences between groups by Wilcoxon rank-sum tests with Holm-Bonferroni adjustment are indicated. * represents groups with statistically significant different flow reduction rates. (**e**) Kaplan-Meier survival curves for mice treated using standard PDT defined by tertiles of flow reduction rate (*n* = 10–11 mice per group). *P* < 0.001 for global Gehan Wilcoxon test of differences between irradiance levels; *p* < 0.001 for the 25.5–100% rBFmin^−1^ group versus each lower group and *p* = 0.027 for 0–12.4% rBFmin^−1^ versus 12.4–25.5% rBFmin^−1^.

**Figure 3 cancers-12-01584-f003:**
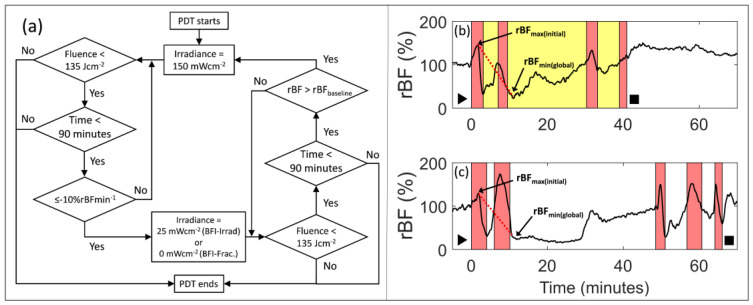
(**a**) Flow chart of the process for blood-flow-informed (BFI) light delivery. Irradiances are either 150 mWcm^−2^ or 25 mWcm^−2^ for BFI-Irrad PDT; and 150 mWcm^−2^ or 0 mWcm^−2^ for BFI-Frac photodynamic therapy (PDT). Representative blood flow traces for (**b**) blood-flow-informed irradiance light delivery (BFI-Irrad) and (**c**) blood-flow-informed fractionated light delivery (BFI-Frac) of radiation-induced fibrosarcoma (RIF) tumors. Light delivery is shaded in red or yellow for illumination at 150 mWcm^−2^ or 25 mWcm^−2^, respectively. ► and ■ indicate the initiation and completion of light delivery, respectively. rBF_max(initial)_ and rBF_min(global)_ are the respective first peak and global minimum of tumor blood flow during light delivery. Dashed red lines in each plot represents the slope of the decrease in rBF between rBF_max(initial)_ and the rBF_min(global)_, described as the overall flow reduction rate.

**Figure 4 cancers-12-01584-f004:**
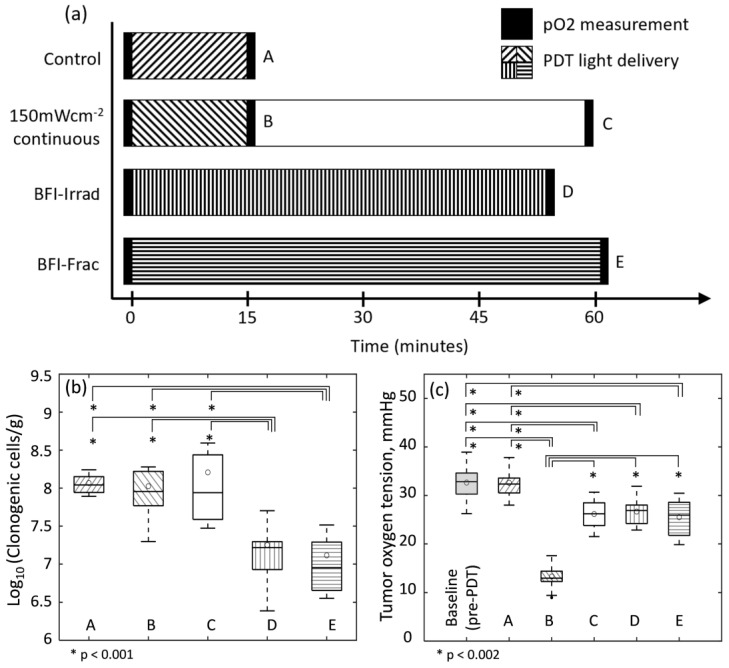
Tumor clonogenicity and oxygenation for standard photodynamic therapy (PDT) with 150 mWcm^−2^-continuous or BFI-PDT (BFI-Irradiance or BFI-Fractionated). (**a**) Timeline of tumor oxygen tension (pO_2_) measurement and excision for clonogenic assay. Dark bands indicate periods of 1-min phosphorescence lifetime measurements of tumor pO_2_; labels A-E indicate timepoints at which mice were euthanized for tumor excision. (**b**) Tumor clonogenicity for each treatment condition/timepoint, 6–7 mice per group. Controls (A) received 15 min of illumination with 150 mWcm^−2^-continuous in the absence of photosensitizer administration; this value was similar to that found for tumors unexposed to light and photosensitizer (log transformed value of 7.9 (0.87 ± 0.3 × 10^8^) clonogenic cells/g) (**c**) Tumor oxygenation for each treatment condition/timepoint, 5 mice per group. Baseline represents the overall pre-PDT tumor pO_2_ for all mice in conditions A-E (*n* = 20 mice). A-E are post-PDT tumor pO_2_ for each control or treatment scheme. For mice that received 150 mWcm^−2^-continuous illumination, pO_2_ measurements were taken twice at different timepoints, B and C, post-PDT as indicated in (**a**). Statistical differences in log-transformed tumor clonogenicity were assessed using a one-way ANOVA and in tumor oxygenation by a mixed effects model to account for repeated measurements on the same animal. The mean for each dataset is indicated by open circles. * represents groups with statistically significant difference.

**Figure 5 cancers-12-01584-f005:**
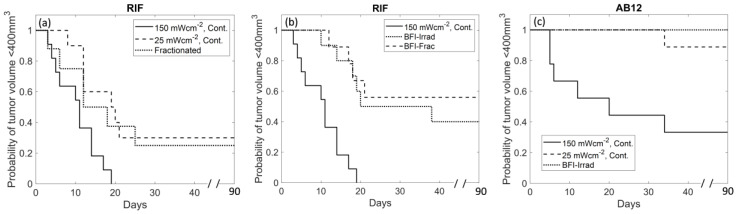
Kaplan Meier survival curves for radiation-induced fibrosarcoma (RIF)-bearing mice treated using (**a**) 150 mWcm^−2^-continuous, 25 mWcm^−2^-continuous, and standard 150 mWcm^−2^-fractionated illumination; (**b**) BFI-Irradiance and BFI-Fractionated illumination, and for comparison purposes, 150 mWcm^−2^-continuous is repeated as a solid line on panel b. (**c**) Kaplan Meier survival curves for murine mesothelioma tumors (AB12) treated using 150 mWcm^−2^-continuous, 25 mWcm^−2^-continuous and BFI-Irradiance illumination. *n* = 9–12 mice per group. Differences in the time-to-400 mm^3^ among groups were assessed using Gehan-Wilcoxon tests for comparisons to 150 mWcm^−2^-continuous for RIF tumors treated with (a) 25 mWcm^−2^-continuous (*p* = 0.031) and standard fractionated (*p* = 0.048) or (b) BFI-Irrad (*p* = 0.006) and BFI-Frac (*p* < 0.002), and for AB12 tumors (**c**) treated with 25 mWcm^−2^-continuous (*p* = 0.021) and BFI-Irrad (*p* = 0.007).

**Table 1 cancers-12-01584-t001:** For RIF tumors, flow reduction rate (overall flow reduction rate for BFI-Irrad and BFI-Frac), treatment length (segregated by time spent at each irradiance), and ΔrBF at 1 h after illumination with standard or BFI-PDT.

Type of PDT	Group	Flow Reduction Rate (%rBFmin^−1^) Median (IQR)	Treatment Length in mMminutes Median (IQR)				ΔrBF(%) at 1 h after PDT Mean (SD)
			150 mWcm^−2^	25 mWcm^−2^	0 mWcm^−2^	Total	
Standard	150 mWcm**^−2^** continuous	23.0 (21.4, 49.8)	15	0	0	15	24.8 (67.3)
	25 mWcm^−**2**^ continuous	9.3 (7.8, 11.6) (*p* < 0.001^1^)	0	90	0	90	−46.2 (23.7) (*p* = 0.001 ^1^)
	150 mWcm**^−2^** fractionated	25.0 (14.9 34,6) (*p* = 0.002^2^)	15	0	15	30	−32.2 (31.3) (*p* = 0.018 ^1^)
Blood-flow informed	BFI-Irrad	5.2 (4.6, 5.4) (*p* < 0.001 ^1,3^, *p* = 0.001^2^)	8 (7, 9)	45 (39, 50)	0	53 (48, 57)	−31.4 (19.7) (*p* = 0.016 ^1^)
	BFI-Frac	10.0 (6.7, 10.2) (*p* < 0.001 ^1,3^)	15	0	46 (34, 55)	61 (49, 70)	−43.6 (29.0) (*p* = 0.002 ^1^)

All *p*-values for pairwise comparisons are adjusted using Tukey’s Honest Significant Difference in determinations ^1^ for comparison to 150 mWcm^−2^-continuous; ^2^ for comparison to 25 mWcm^−2^-continuous; ^3^ for comparison to 150 mWcm^−2^-fractionated. RIF: radiation-induced fibrosarcoma; PDT: photodynamic therapy; rBF: relative blood flow; BFI-Irrad: blood-flow-informed irradiance; BFI-Frac: blood-flow-informed fractionated; IQR: interquartile range; SD: standard deviation.

**Table 2 cancers-12-01584-t002:** For AB12 tumors, flow reduction rate (overall flow reduction rate for BFI-Irrad), treatment length (segregated by time spent at each irradiance), and ΔrBF at 1 h after PDT with 150 mWcm^−2^-continuous, 25 mWcm^−2^-continuous or BFI-Irrad.

Group	Flow Reduction Rate (%rBFmin^−1^) Median (IQR)	Treatment Length in Minutes Median (IQR)				ΔrBF(%) at 1 h after PDT Mean (SD)
		150 mWcm^−2^	25 mWcm^−2^	0 mWcm^−2^	Total	
150 mWcm^−^^2^-continuous	20.7 (15.5, 32.6)	15	0	0	15	−10.7 (21.6)
25 mWcm^−^^2^-continuous	13.4 (11.9, 16.8)	0	90	0	90	−56.6 (18.6) (*p* = 0.001 ^1^)
BFI-Irrad	6.0 (3.7, 12.1) (*p* = 0.004 ^1^, *p* = 0.04 ^2^)	11 (9, 14)	27 (9, 36)	0	37 (23, 45)	−58.5 (15.3) (*p* = 0.001 ^1^)

All *p*-values for pairwise comparisons are adjusted using Tukey’s Honest Significant Difference in determinations ^1^ for comparison to 150 mWcm^−2^-continuous; ^2^ for comparison to 25 mWcm^−2^-continuous. AB12: mouse mesothelioma cell line; PDT: photodynamic therapy; rBF: relative blood flow; BFI-Irrad: blood-flow-informed irradiance; BFI-Frac: blood-flow-informed fractionated; IQR: interquartile range; SD: standard deviation.

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
