# Peer review of "Blood Flow Measurements Enable Optimization of Light Delivery for Personalized Photodynamic Therapy"

_cancers, 2020, doi:10.3390/cancers12061584_

Round 1

Reviewer 1 Report

This study reports the development of blood-flow-informed PDT to improve treatment outcomes and reduce treatment duration. The relative blood flow in the tumor is monitored by non-invasive diffuse correlation spectroscopy (DCS). Such approach, permits continuous monitoring during PDT without interfering with treatment light delivery, is different from existing fluorescence image-guided PDT. The proof-of-concept manuscript is of high scientific merit and well written, and therefore, it is recommended for a publication after minor elaborations and adjustments to the discussion of the results. The use of timelines and a flow chart to described experimental plans is a strength.

1. Photofrin was injected 20-24 hours before illumination. The authors should clarify why this photosensitizer-light interval was selected, and the location of Photofrin in the tumor (in vessels, cells, or both) during the illumination time.

2. The authors should acknowledge that external beam PDT induces no to minimal tissue heating (2–3 °C) at irradiances ≤150 mW/cm2. (Shafirstein et al. BJC, 2018)

3. In figure 3, it is unclear why/how a cutoff value of 10% per minute was selected. Discussions on 'how changing the cutoff value impacts treatment outcome' will strengthen the manuscript.

4. The penetration depth of DSC should be discussed. Does DCS measure the blood flow of the entire tumor or only the rim of the tumor?

5. ΔrBF was used to represent the change in vascular function between two timepoints. The authors suggested that BFI-PDT leads to more 'vascular damage', compared to using 150 mWcm-2-continuous PDT. Vascular damage can imply endothelial dysfunction, permeabilization, or shutdown. What assays have been used to confirm 'vascular damage', besides ΔrBF?

6. BFI-Irrad induced vascular shutdown at 1 hr post PDT could likely restrict the subsequent delivery of chemo/biological agents. Any potential benefits of high-fluence PDT induced vascular permeabilization (in combination with chemo/biological agents) should also be discussed.

Reviewer 2 Report

Presented manuscript from a lab of an established investigator describes studies of possible improvement of photodynamic therapy outcome by adjusting light fluence to the changes in tumor blood flow (oxygenation, nutrient delivery). Presented results are sound and conclusions formulated carefully. The reviewer has no issues with presented manuscript.

Author Response

We would like to thank reviewer for the positive comment!

Reviewer 3 Report

The present manuscript describes the use of diffuse correlation spectroscopy (DSC) to measure in real time, and non-invasively, the blood flow of illuminated tumors. The paper is well written and the technical approaches are suited according to the paper goals. DSC was used to compare different PDT protocols namely: standard PDT protocols with high, low and fractionated fluence rates as well as the blood-flow-informed (BFI) protocols developed by the authors.  The authors concluded that high fluence rates (150 mW/cm2) mediated poor therapeutic outcomes owing to oxygen deprivation which is indeed well known in field. It is also shown that BFI, using high fluence rates (150 mW/cm2), enabled better therapeutic outcomes when compared to the continuous protocol using 150 mW/cm2. My main concern relies in the poor advantage of using such intricate BFI protocols when compared with a standard low fluence rate protocol (25 mW/cm2). The authors should provide information regarding the statistical significance between the BFI protocols and the 25 mW/cm2 continuous protocol (Figure 5 b and c).

Author Response

Statistical significance between BFI protocols and the 25mWcm-2-continuous protocol are provided in Lines 433-434 and Lines 443-444.

Although Kaplan Meier survival estimates for BFI protocols are not statistically better than that of 25mWcm-2-continuous, we respectfully disagree with reviewer that BFI-PDT has poor advantage over standard low fluence rate protocol for the following reasons.

(1) For RIF model, both BFI protocols achieve better median complete response rate and longer median time-to-400mm3 (BFI-Irrad:40%, 29 days ; BFI-Frac:50%, 39 days) than standard 25mWcm-2-continuous (30%, 20 days) (Table S1). Similarly for AB12 model, BFI-Irrad achieves better complete response rate (100%) compared to standard 25mWcm-2-continuous (89%). (Table S2).  Thus, the potential exists that with further optimization of the threshold value (see response to comment 3 of Reviewer 1), BFI protocols will achieve better response than those of low fluence rate.

(2) BFI protocols were able to provide therapeutic response that was similar to that of 25mWcm-2-continuous in much shorter treatment time. For BFI-Irrad, treatment time was ~41% shorter than that required for 25mWcm-2-continuous; while for BFI-Frac treatment time was ~32% shorter. This can be huge advantage over standard low fluence rate protocol that requires clinically undesirable lengthy treatment time.

We emphasize this important time advantage of BFI light delivery in the Conclusion with the following statement:

“BFI-PDT provided the desired benefit of achieving a therapeutic response that was similar to that of low irradiance PDT but required a much shorter treatment time.”